# Evaluating Adversarial Robustness of Low dose CT Recovery

**Kanchana Vaishnavi Gandikota**            KANCHANA.GANDIKOTA@UNI-SIEGEN.DE
**Paramanand Chandramouli**            PARAMANAND.CHANDRAMOULI@UNI-SIEGEN.DE
**Hannah Droege**            HANNAH.DROEGE@UNI-SIEGEN.DE
**Michael Moeller**            MICHAEL.MOELLER@UNI-SIEGEN.DE
*Department of Computer Science*
*University of Siegen*

## Abstract

Low dose computer tomography (CT) acquisition using reduced radiation or sparse angle measurements is recommended to decrease the harmful effects of X-ray radiation. Recent works successfully apply deep networks to the problem of low dose CT recovery on benchmark datasets. However, their robustness needs a thorough evaluation before use in clinical settings. In this work, we evaluate the robustness of different deep learning approaches and classical methods for CT recovery. We show that deep networks, including model based networks encouraging data consistency are more susceptible to untargeted attacks. Surprisingly, we observe that data consistency is not heavily affected even for these poor quality reconstructions, motivating the need for better regularization for the networks. We demonstrate the feasibility of universal attacks and study attack transferability across different methods. We analyze robustness to attacks causing localized changes in clinically relevant regions. Both classical approaches and deep networks are affected by such attacks leading to change in visual appearance of localized lesions, for extremely small perturbations. As the resulting reconstructions have high data consistency with original measurements, these localized attacks can be used to explore the solution space of CT recovery problem.
**Keywords:** Computer tomography, robustness, adversarial attacks, image reconstruction.

## 1. Introduction

Computer tomography (CT) is a non-invasive imaging technique widely used in medical diagnosis. The procedure involves recording attenuated X-ray radiation projected at different angles by a scanner rotating around a target. The recorded measurements are arranged into a sinogram, from which a CT image is reconstructed. As exposure of patients to X-rays poses serious health risks, different solutions to low-dose CT acquisition have been proposed under the ALARA (as low as reasonably achievable) principle (Slovis, 2002; Newman and Callahan, 2011). These protocols can be broadly classified into two categories- i) adjusting the settings on the CT scanner tube to reduce total number of X-ray photons ii) recording measurements from fewer projection angles. However, there exists a trade-off between dose reduction during CT acquisition and diagnostic quality. Lower number of X-ray photons degrades reconstruction quality due to increased image noise level. On the other hand, CT recovery from fewer projection angles can suffer from severe artefacts. Further, sparse-view CT is an ill-posed problem, and there can be many valid solutions for the same measurement.

Traditional approaches to ill-posed CT recovery impose suitable priors such as total variation (Sidky et al., 2006; Chen et al., 2013) in a variational reconstruction algorithm.

Recent works (Chen et al., 2017; He et al., 2020) train deep networks for sparse view CT recovery. While deep networks achieve impressive performance, they lack convergence guarantees provided by classical approaches. Moreover, sensitivity of deep networks to adversarial examples (Szegedy et al., 2014; Goodfellow et al., 2015) is a serious concern in clinical applications. In this paper, we analyze the robustness of different classical and deep learning methods to norm bounded additive adversarial perturbations. We show that deep networks, including the model inspired ones are significantly more susceptible to untargeted adversarial examples than classical methods. Surprisingly, even the poor reconstructions display a reasonable data consistency with their input, motivating the need for better regularization in these networks. We demonstrate the feasibility of universal attacks and study attack transferability across different methods. We also show that both classical and deep learning methods are sensitive to localized adversarial attacks aiming to alter the visual appearance of a small diagnostically relevant regions. Such local attacks are possible with very low adversarial noise and high data consistency with original measurement, indicating that multiple diagnostically different solutions can be obtained with high data consistency.

## 2. Background and Related Work

### 2.1. CT Acquisition and Reconstruction

In CT acquisition, the forward operator is given by the 2D Radon transform (Radon, 1986) which models the attenuation of the radiation passing through the target by calculating line integral along the path of X-ray beam. The measurement, which is a sinogram consists of the recorded integrals for different distances and measurement angles. Since the Radon transform is linear, the measurement process can be written as:

$$f = Au + n \tag{1}$$

where $f$, $A$, $u$, $n$ represent the sinogram, forward Radon transform, ground truth image and measurement noise respectively. The aim is to recover a CT image $\hat{u}$ from the sinogram $f$. Linearly filtering in Fourier space, commonly referred to as filtered back projection (FBP) (Feldkamp et al., 1984), is one standard classical approach to CT recovery. Variational approaches (Sidky et al., 2006; Chen et al., 2013) find a minimizer of the energy function

$$\hat{u} = \arg\min_u \frac{1}{2}\|Au - f\|^2 + R(u) \tag{2}$$

for a suitable regularizer $R(u)$ such as the total variation $\|\nabla u\|_{2,1}$. In the following, we review recent deep learning approaches for such ill-posed image recovery problems.

### 2.2. Deep learning for Image and CT Reconstruction

Deep learning approaches to image reconstruction tasks encompass a wide array of methods: *i) Fully learned methods* directly invert the forward imaging model (Zhu et al., 2018). Examples for CT recovery include iRadonmap (He et al., 2020) and ADAPTIVE-Net (Ge et al., 2020), which also learn the filtered back projection operation

$$\hat{u} = \mathcal{N}(f) \tag{3}$$

*ii) Learning deep neural network post-processors* denoise an initial reconstruction such as output from the filtered-back-projection operator $B^{\dagger}(\cdot)$ (Chen et al., 2017; Jin et al., 2017; Yang et al., 2018; Zhang et al., 2018; Pelt et al., 2018; Kuanar et al., 2019)

$$\hat{u} = \mathcal{N}(B^{\dagger}(f)) \tag{4}$$

*iii) Unrolled optimization networks* are end-to-end trained model inspired neural networks which unroll fixed iterations of algorithms such as gradient descent, primal-dual hybrid gradient, projected gradient descent with learned parameters (Adler and Öktem, 2017; Aggarwal et al., 2018; Adler and Öktem, 2018). Closely related is the method of using trained networks for projection or proximal step (He et al., 2018; Gupta et al., 2018).

*iv) Use of trained/untrained neural network priors in a variational inference* (Bora et al., 2017; Rick Chang et al., 2017; Meinhardt et al., 2017; Ulyanov et al., 2018; Heckel et al., 2019). For CT recovery, (Baguer et al., 2020) use untrained neural network prior (Ulyanov et al., 2018), and (Song et al., 2022) use generative models trained on CT images.

In this work, we analyze the adversarial robustness of the deep learning paradigms $i) - iii)$, which can recover CT images in a single forward pass. In addition, we consider the classical approaches of filtered back projection and energy minimization with TV prior. We exclude $iv)$ in our experiments due to high computational complexity.

## 2.3. Adversarial Attacks on Image Reconstruction

Adversarial attacks refer to a phenomenon where a carefully crafted imperceptible change in the input causes a catastrophic failure of neural networks. Starting from (Szegedy et al., 2014; Goodfellow et al., 2015), many works focused on stronger adversarial attacks and mechanisms to defend classification networks from adversarial attacks. Recent works (Antun et al., 2020; Raj et al., 2020) demonstrated the susceptibility of image reconstruction networks to adversarial attacks. While (Antun et al., 2020) investigate instabilities to perturbations in the image domain, (Raj et al., 2020) consider adversarial examples in measurement domain and propose adversarial training to improve robustness. However, these works consider mainly untargeted attacks for networks doing direct inversion or post-processing. A few recent works (Choi et al., 2019; Gandikota et al., 2022) also investigated the adversarial robustness of image restoration methods. (Cheng et al., 2020) show that MRI recovery networks can fail to recover tiny features under adversarial attacks and perform robust training to increase the network's sensitivity to these small features. (Darestani et al., 2021; Morshuis et al., 2022) show that adversarial perturbations can alter diagnostically relevant regions in recovered MRI images. In the context of CT recovery, (Huang et al., 2018) perform preliminary investigations whether additive adversarial perturbations can lead to incorrect reconstruction of an existing lesion. Closely related to our work, (Genzel et al., 2022; Wu et al., 2022) also investigate the adversarial robustness of different approaches for CT recovery. They mainly considered untargeted attacks, with some preliminary experiments in (Genzel et al., 2022) on targeted changes indicating that reconstruction networks are largely robust to targeted changes. In contrast, we study susceptibility of CT recovery methods to untargeted attacks, universal attacks and localized adversarial attacks targeting diagnostically relevant regions in thoracic CT scans from (Armato III et al., 2011).

## 3. Analyzing Stability of (CT) Image Recovery

Ideally, the recovery algorithm or network $\mathcal{N}$ should have a small Lipschitz constant $L$ so that small changes in the input produce only small bounded changes in the reconstruction,

$$\|\mathcal{N}(f_1) - \mathcal{N}(f_2)\| \leq L\|f_1 - f_2\|. \tag{5}$$

However, exactly computing Lipschitz constant has extremely high computational complexity (Jordan and Dimakis, 2020) even for moderately sized neural networks. Recent works (Combettes and Pesquet, 2020; Jordan and Dimakis, 2020; Huang et al., 2021) instead estimate an upper bound on Lipschitz constant. On the other hand, it is easier to analyze the stability of classical approaches. The stability of the standard linear techniques can be analyzed via the singular values of the reconstruction operator, see, e.g. (Bauermeister et al., 2020) for learning linear reconstructions in such a context. For nonlinear variational energy minimization approaches, a stability estimate shown in (Burger et al., 2007) is

$$\|f_1 - f_2\|^2 \geq \|Au_1 - Au_2\|^2 + 2\langle p_1 - p_2, u_1 - u_2 \rangle, \quad p_1 \in \partial R(u_1), \ p_2 \in \partial R(u_2), \tag{6}$$

where the term $\langle p_1 - p_2, u_1 - u_2 \rangle$ is the 'symmetric Bregman distance' with respect to the convex regularizer $R$.

### 3.1. Adversarial Attacks on CT recovery

Adversarial attacks on image recovery methods make small changes to the inputs causing unpredictable large changes in the output. In this work, we consider robustness to tiny $L_\infty$ norm bounded additive perturbations in the measurement domain. We assume that the parameters of the neural network $\mathcal{N}$ or the recovery algorithm is fully known to the attacker. **Untargeted Attacks:** Here the aim is to find an additive $L_\infty$ norm constrained perturbation in the measurement domain that maximizes the reconstruction error:

$$\delta_{adv} = \operatorname*{argmax}_{\delta \in \mathrm{R}^m} \|\mathcal{N}(f + \delta) - \mathcal{N}(f)\|_2 \text{ s.t. } \|\delta\|_\infty \leq \epsilon. \tag{7}$$

**Localized Attacks:** Here the goal is to find an additive $L_\infty$ norm constrained perturbation that produces a change the visual appearance to alter predicted malignancy in a localized clinically relevant region. We utilize an adversarially trained classifier $\mathcal{N}_\theta$ trained to classify chest CT nodules to guide the attack towards a plausible change in visual appearance locally. Note that using a non-robust classifier in the attack would cause mis-classification even without perceptible changes in reconstruction. Our localized attack can be formulated as:

$$\delta_{adv} = \operatorname*{argmax}_{\delta \in \mathrm{R}^m} E(\mathcal{N}_\theta \left( g_c \left( \mathcal{N}(f + \delta) \right) \right), y) \text{ s.t. } \|\delta\|_\infty \leq \epsilon. \tag{8}$$

where $g_c(\cdot)$ crops the region of interest, and $y = \mathcal{N}_\theta \left( g_c \left( \mathcal{N}(f) \right) \right)$ is the predicted label for the region of interest in the clean reconstruction. $E(\cdot)$ refers to the energy function (loss) to be maximized for binary classification of nodules, which is the binary cross entropy loss. To ensure that the degradation remains localized, and to avoid artifacts at the boundary of the local region, we apply a smoothed mask to the adversarial noise setting at every step. The mask is calculated as the sinogram of the Gaussian smoothed spatial mask corresponding to region of interest, and normalized to have maximum value of 1.

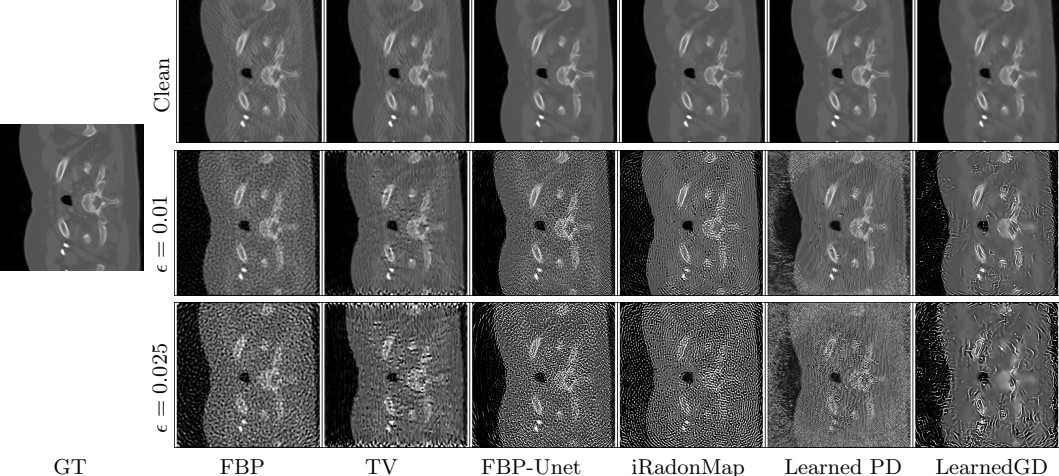

Figure 1: Untargeted attack on CT reconstruction methods for $\epsilon$ values 0.01 and 0.025.

| Method | $\hat{u}$ PSNR/SSIM/$d_{Breg}$ | $(A\hat{u}, f)$ PSNR | $\epsilon$ | $\hat{u}_\delta$ PSNR/SSIM/$d_{Breg}$ | $(A\hat{u}_\delta, f)$ PSNR | $(A\hat{u}_\delta, f_\delta)$ PSNR | $(f, f_\delta)$ PSNR | $L_b$ Empir |
|---|---|---|---|---|---|---|---|---|
| FBP | 30.37/0.738/0.018 | 33.82 | 0.01 | 25.18/0.448/0.029 | 33.36 | 33.37 | 40.20 | |
| | | | 0.025 | 18.68/0.194/0.049 | 31.47 | 31.43 | 32.51 | 15.03 |
| | | | 0.05 | 13.02/0.074/0.081 | 28.46 | 28.34 | 26.91 | |
| TV | 31.62/0.763/0.018 | 36.52 | 0.01 | 25.20/0.615/0.026 | 35.62 | 35.72 | 40.36 | |
| | | | 0.025 | 18.32/0.365/0.044 | 32.51 | 33.24 | 32.71 | 16.52 |
| | | | 0.05 | 12.99/0.150/0.077 | 28.66 | 30.01 | 27.22 | |
| FBP-Unet | 35.47/0.837/0.013 | 36.47 | 0.01 | 18.39/0.287/0.081 | 35.06 | 35.71 | 40.28 | |
| | | | 0.025 | 12.18/0.095/0.152 | 29.82 | 30.95 | 32.77 | 46.71 |
| | | | 0.05 | 7.38/0.034/0.227 | 24.86 | 25.93 | 27.39 | |
| iRadonMap | 33.94/0.810/0.014 | 36.03 | 0.01 | 17.98/0.326/0.062 | 29.62 | 29.90 | 40.22 | |
| | | | 0.025 | 10.85/0.084/0.140 | 24.07 | 24.51 | 32.60 | 43.80 |
| | | | 0.05 | 6.24/0.026/0.215 | 21.50 | 21.98 | 27.16 | |
| LearnedPD | 35.73/0.842/0.012 | 36.46 | 0.01 | 9.47/0.164/0.230 | 25.27 | 25.50 | 40.48 | |
| | | | 0.025 | 3.38/0.030/0.467 | 23.05 | 23.38 | 32.95 | 143.39 |
| | | | 0.05 | 0.36/ 0.008/0.623 | 28.28 | 28.72 | 27.17 | |
| LearnedGD | 34.55/0.815/0.014 | 36.43 | 0.01 | 21.14/0.504/0.036 | 35.18 | 35.62 | 40.39 | |
| | | | 0.025 | 13.90/0.291/0.069 | 31.62 | 32.82 | 32.80 | 30.48 |
| | | | 0.05 | 8.64/0.180/0.099 | 28.11 | 29.64 | 27.50 | |

Table 1: Comparison of robustness to untargeted attacks on different CT reconstruction methods using 20 attack iterations on first 100 samples LoDoPAB testset.

**Universal Attacks:** Here we aim to find an input-agnostic $L_\infty$ norm constrained adversarial perturbation that maximizes the reconstruction error of a recovery method $\mathcal{N}$ for any input.

$$\delta_{uniadv} = \underset{\delta \in \mathrm{R}^m}{\mathrm{argmax}} \sum_{\text{examples i}} \|\mathcal{N}(f_i + \delta) - \mathcal{N}(f_i)\|_2 \text{ s.t. } \|\delta\|_\infty \leq \epsilon. \tag{9}$$

We solve the constrained optimization problems (7), (8) and (9) using projected gradient descent (PGD) (Madry et al., 2018), with gradient updates using Adam (Kingma and Ba, 2015).

## 4. Experiments and Results

We conduct experiments with low-dose parallel beam (LoDoPaB) CT dataset (Leuschner et al., 2021), consisting of data pairs of simulated low-intensity measurements for sampling 513 out of 1000 parallel beams and corresponding ground truth human chest CT images from the LIDC/IDRI dataset(Armato III et al., 2011). We evaluate the robustness of the following approaches: i) Filtered back projection(FBP) ii) FBP-Unet (Chen et al., 2017) post-processing FBP outputs, iii) iRadonmap(He et al., 2020), which also learns back projection in addition to pre-processing, iv) LearnedGD, learned gradient descent v) Learned Primal Dual(Adler and Öktem, 2018) vi) Total Variation regularization. For the localized attacks, we obtain the locations of regions of interest corresponding to ground truth from the LIDC-IDRI dataset. For malignancy classification, we use a BasicResNet model(Al-Shabi et al., 2019) adversarially trained on nodule patches from LIDC-IDRI dataset. We consider additive perturbations are $L_\infty$ norm bounded by 1%, 2.5% and 5% of the intensity range of the clean observation. Further experiment details are found in Appendix A. We will make the code for our experiments publicly available up on acceptance. In the following $f$, $f_\delta$, $\hat{u}$ and $\hat{u}_\delta$ denote the clean and adversarial sinogram measurements and the corresponding recovered CT images respectively.

**Performance Metrics:** We measure the PSNR, SSIM and the TV Bregman distance of the reconstructions with clean and adversarial inputs with respect to the ground truth (setting the corresponding subgradient to zero if the norm of the gradient is below a threshold of $10^{-5}$, which we consider to be 'numerically zero'). We also measure data consistency of the reconstructions with respect to the clean and adversarial sinograms in terms of PSNR. Further, we empirically compute a lower bound for Lipschitz constant of each method as

$$L_b(\mathcal{N}) = \left( \frac{\|\mathcal{N}(f_\delta) - \mathcal{N}(f)\|}{\|\delta\|} \right)_{\max}$$

which is the maximum value obtained across the test set of 100 CT images for the three adversarial noise levels with 5 random restarts (a total of 1500 examples). For localized attacks, we additionally compare the PSNR values in the local region, and the region exterior to it, for reconstructions with clean and adversarial inputs.

**Untargeted Attacks:** Table 1 and Figure 1 illustrate the results of untargeted attacks (7). The results demonstrate that in absence of adversarial noise, the neural network approaches provide qualitatively better reconstructions than FBP and TV minimization. However, their reconstructions are also more susceptible to adversarial perturbations despite training with inputs corrupted by Poisson noise. Among the deep learning approaches, the learned primal-dual network which provides the best reconstructions from clean inputs is also the most unstable to perturbations, where as the learned gradient descent is more stable. This is also reflected in the empirical Lipschitz lower bound which is the highest for LearnedPD. This high sensitivity to adversarial attacks is surprising as LearnedPD also encourages data consistency in its (fixed number of) iterations. Among the classical methods, FBP and TV minimization have similar stability in terms of PSNR and $L_b$, while TV is better in terms of SSIM and Bregman distance as one would have hoped considering the provable stability (6). Interestingly, the adversarial perturbations do not heavily affect the data consistency of the recovered CT images for all the methods. The adversarially affected CT

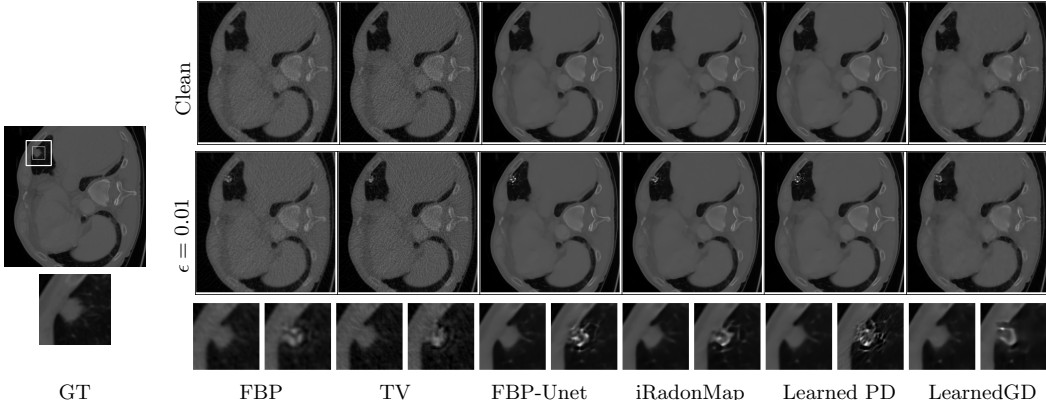

Figure 2: Localized attack on CT recovery for $\epsilon = 0.01$. For each method, the third row shows the cropped patches from the clean (left) and adversarial (right) reconstructions.

| | Optimized | | | | | Unseen | | | | |
|---|---|---|---|---|---|---|---|---|---|---|
| | FBP | FBP-Unet | iRadonMap | LearnedGD | LearnedPD | FBP | FBP-Unet | iRadonMap | LearnedGD | LearnedPD |
| Clean | 30.37/0.738 | 35.47/0.837 | 33.95/0.810 | 34.55/0.815 | 35.73/0.843 | 30.53/0.714 | 35.67/0.824 | 34.19/0.799 | 34.74/ 0.802 | 35.92/0.829 |
| $\epsilon = 0.01$ | 22.87/0.340 | 17.96/0.223 | 15.58/0.263 | 18.41/0.542 | 7.19/0.139 | 23.27/ 0.337 | 18.59/0.225 | 16.29/ 0.262 | 19.04/0.538 | 7.93/0.161 |
| $\epsilon = 0.05$ | 9.87/0.036 | 4.49/0.023 | 3.303/0.011 | 3.80/0.179 | -3.71/0.003 | 10.34/ 0.036 | 4.95/0.022 | 3.82/0.0108 | 4.32/0.183 | -2.95/0.003 |

Table 2: Universal adversarial attack on CT recovery. PSNR/SSIM values for clean samples and samples affected by additive universal perturbation are shown.

reconstructions from LearnedPD with an extremely low average PSNR (0.36 dB) still have a good data consistency (28.7 dB) with the input measurement, showing instabilities typical to unregularized solutions to the recovery problem. Results of similar untargeted attack on LoDoPAB_200 dataset are provided in Table 6 of the appendix.

**Universal Attacks:** We perform input-agnostic attack universal attack (9) by optimizing over a set of 100 samples. Table 2 shows the effect of this adversarial perturbation on the optimized examples and its generalizability on a different 100 examples not seen during optimization, indicating that CT recovery methods can also be affected by universal attacks.

**Transferability of Adversarial Examples:** In context of image classification, adversarial perturbations are often transferable across different networks (Liu et al., 2017). Even for CT recovery, we find that the adversarial perturbations and even universal perturbations transfer across different methods, detailed results are provided in the Tables 4 & 5 of appendix.

**Localized Attacks:** Table 3 and Figure 2 provide the results of our experiments with localized attacks (8) on different CT recovery methods. Sample reconstructions from different methods with clean and adversarial inputs are compared in Figure 2. The results clearly demonstrate visible alteration in the region of interest $\hat{u}_i$ indicated by the inner square marked in the ground truth image. Our attack successfully achieves this modification, barely affecting the reconstruction in the exterior region $\hat{u}_e$.

| Method | $\hat{u}$ PSNR/SSIM | $\hat{u}_i\|\hat{u}_e$ PSNR | $(A\hat{u},f)$ PSNR | $\epsilon$ | $\hat{u}_\delta$ PSNR/SSIM | $\hat{u}_{\delta_i}\|\hat{u}_{\delta_e}$ PSNR | $(A\hat{u}_\delta,f)$ PSNR | $(A\hat{u}_\delta,f_\delta)$ PSNR | $(f,f_\delta)$ PSNR | success rate |
|---|---|---|---|---|---|---|---|---|---|---|
| FBP | 30.86/0.787 | 31.45\|30.86 | 33.81 | 0.01 | 30.60/0.782 | 22.29\|30.83 | 33.79 | 33.77 | 55.09 | 100 |
| | | | | 0.025 | 30.35/0.772 | 20.93\|30.67 | 33.75 | 33.42 | 47.55 | 100 |
| | | | | 0.05 | 29.97/0.751 | 19.89\|30.34 | 33.70 | 32.63 | 41.08 | 100 |
| TV | 32.36/0.829 | 31.84\|32.37 | 36.52 | 0.01 | 32.00/0.825 | 22.70\|32.32 | 36.48 | 36.42 | 54.77 | 100 |
| | | | | 0.025 | 31.62/0.812 | 21.26\|32.07 | 36.46 | 35.66 | 46.97 | 100 |
| | | | | 0.05 | 30.65/0.767 | 20.28\|31.15 | 36.35 | 33.59 | 40.11 | 100 |
| FBP-Unet | 36.94/0.909 | 35.67\|36.95 | 36.50 | 0.01 | 34.85/0.902 | 19.43\|36.61 | 36.46 | 36.43 | 55.11 | 100 |
| | | | | 0.025 | 33.79/0.889 | 17.82\|35.87 | 36.37 | 35.84 | 47.83 | 100 |
| | | | | 0.05 | 33.15/0.877 | 17.27\|35.11 | 36.11 | 34.42 | 41.90 | 100 |
| iRadonMap | 35.25/0.888 | 34.07\|35.27 | 36.09 | 0.01 | 33.70/0.883 | 18.85\|35.12 | 36.03 | 36.03 | 55.32 | 100 |
| | | | | 0.025 | 32.68/0.875 | 16.53\|34.76 | 35.95 | 35.52 | 48.08 | 100 |
| | | | | 0.05 | 30.60/0.808 | 15.32\|32.73 | 35.55 | 33.39 | 40.81 | 100 |
| LearnedPD | 37.22/0.913 | 35.97\|37.23 | 36.49 | 0.01 | 33.15/0.854 | 18.34\|35.08 | 36.28 | 36.10 | 53.74 | 100 |
| | | | | 0.025 | 29.90/0.753 | 16.15\|31.57 | 35.33 | 34.57 | 45.41 | 100 |
| | | | | 0.05 | 25.05/0.559 | 14.52\|25.72 | 33.29 | 31.74 | 38.41 | 100 |
| LearnedGD | 35.80/0.891 | 34.86\|35.82 | 36.49 | 0.01 | 34.86/0.886 | 22.02\|35.71 | 36.46 | 36.42 | 55.29 | 100 |
| | | | | 0.025 | 34.49/0.883 | 20.98\|35.53 | 36.42 | 35.99 | 48.41 | 100 |
| | | | | 0.05 | 34.12/0.875 | 21.11\|35.04 | 36.28 | 34.72 | 42.44 | 100 |

Table 3: Comparison of robustness to localized attacks on different CT reconstruction method evaluated on 100 samples LoDoPAB testset.

Table 3 summarizes our results for localized attacks for three levels of adversarial noise. The subscripts $i$ and $e$ denote the restriction to the interior and exterior of the local region to be attacked. Due to masking, the magnitudes of additive perturbation are extremely small, with high PSNR values between the clean and adversarial inputs for all noise levels. Still, our attack is almost always successful in producing local degradations that change the malignancy prediction. This is also reflected in the steep PSNR drop in the local region $\hat{u}_i$, while the PSNR in the exterior region are mostly unaffected. While the classical approaches are more robust to untargeted attacks, they are also sensitive to local changes. This is a direct consequence of ill-posedness of the recovery problem, as we observe nearly similar data consistency of the recovered $\hat{u}_\delta$ with both clean and adversarial inputs. In a recent work (Dröge et al., 2022) demonstrate that the CT images of varying malignancy level can be solutions the same measurement with a high data consistency, but by modifying the reconstruction loss. Our localized attacks also show that the adversarial noise necessary to change the malignancy is extremely small for a variety of methods and the resulting solutions demonstrate high data consistency with both clean and adversarial inputs. One could utilize such attacks beneficially to efficiently explore diagnostically different reconstructions with a very high degree of data consistency with sinogram. This can be used by a medical doctor to choose the most plausible reconstruction in making diagnosis.

## 5. Conclusions

In this work, we analyzed the adversarial robustness of classical and deep learning methods to CT recovery. We showed that deep learning methods are more sensitive to untargeted adversarial examples than the classical approaches. Even model inspired unrolled networks are susceptible to adversarial examples, even though they encourage data consistency within

the network. While the quality of the recovered CT images degrades, we find that the recovered images still exhibit a good degree of data consistency. This motivates the need to improve robustness of deep networks for CT recovery via better regularization techniques or via adversarial training. Further, we demonstrated the susceptibility of CT recovery methods to universal attacks, and showed that perturbations can transfer across different methods. We also find that the classical methods and deep learning methods are similarly affected by adversarial examples targeting small localized regions. Moreover, such attacks are successful for extremely small perturbations already, such that the resulting reconstructions have high data consistency with original measurements. Therefore, the proposed localized attacks could serve as a way to explore the solution space of reconstruction networks.

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

## Appendix A. Experiment Details

In our experiments we analyzed the robustness of the following methods: i) Filtered back projection(FBP) ii) FBP-Unet (Chen et al., 2017), iii) iRadonmap(He et al., 2020), iv) LearnedGD, v) Learned Primal Dual(Adler and Öktem, 2018) vi) Total Variation regularization. For the learned methods ii)-v), we use the pretrained models[1] from (Baguer et al., 2020) trained on the full training set excluding iRadonmap (which we trained ourselves to full convergence). For FBP, we employ the Hann filter with low-pass cut-off of 0.6410, the best setting for this dataset in (Baguer et al., 2020). When attacking FBP-Unet Equation (4), we also backpropagate through $B^{\dagger}(\cdot)$. For TV minimization, we used 500 gradient descent steps, with a TV weight of 1e-3, and the attack backpropagates through all the gradient descent steps. For the localized attacks, we obtain the locations of regions of interest corresponding to ground truth from the LIDC-IDRI dataset (Armato III et al., 2011). We exclude the images where the patch surrounding the nodule does not lie fully with in the central cropped region of LoDoPAB dataset. For malignancy classification, we consider a BasicResNet model(Al-Shabi et al., 2019) trained on nodule patches from LIDC-IDRI dataset. We utilize the adversarially trained model from (Dröge et al., 2022).

**Untargeted Attacks:** We perform untargeted attacks (Equation (7)) using step size of $1e-3$ and 20 PGD steps and choose the best adversarial noise from 5 random restarts.

**Universal Attacks:** We perform universal attack (9) on each method by optimizing a single $L\infty$ norm constrained untargeted adversarial perturbation for hundred examples using step size of $1e-3$ with PGD steps for 500 epochs.

**Localized Attacks:** We perform adversarial attacks effecting localized changes Equation (8)

---

1. https://github.com/oterobaguer/dip-ct-benchmark

| Source Noise | FBP | FBP-Unet | iRadonMap | LearnedGD | LearnedPD |
|---|---|---|---|---|---|
| Clean | 30.37/0.738 | 35.47/0.837 | 33.94/0.810 | 34.55/0.815 | 35.73/0.842 |
| FBP | **18.68/0.194** | 16.19/0.139 | 15.41/0.131 | 16.04/0.138 | 16.19/0.151 |
| FBP-Unet | 22.03/0.325 | **12.19/0.095** | 16.33/0.173 | 17.98/0.279 | 14.10/0.125 |
| iRadonMap | 20.72/0.284 | 15.18/0.152 | **10.86/0.084** | 15.45/0.197 | 16.01/0.171 |
| LearnedGD | 21.17/0.375 | 15.42/0.275 | 15.96/0.271 | **13.90/0.290** | 15.28/0.241 |
| LearnedPD | 26.39/0.553 | 25.33/0.604 | 26.19/0.590 | 26.23/0.603 | **3.38/0.030** |
| TV | 19.19/0.365 | 16.94/0.289 | 16.78/0.305 | 16.66/0.280 | 16.75/0.333 |

Table 4: Evaluating transferability of adversarial noises for $\epsilon$=0.025

using step size of $1e-3$ and iterate for a maximum of 50 PGD steps till the local patch is misclassified. We choose the best adversarial noise from 5 random restarts.

## Appendix B. Transferability of Adversarial Examples

Transferability of adversarial examples is studied in context of image classification networks, to examine the possibility of black box attacks. We investigate the transferability of adversarial examples across different CT recovery methods, i.e. we test whether, an adversarial example crafted for a "source" CT recovery method also reduces the quality of reconstruction of a different target method for CT recovery. Table 4 summarizes the results of transferability for CT recovery methods, for $\epsilon$ value of 0.025. The results demonstrate that the adversarial examples are indeed transferable across different methods to some extent. The adversarial examples for classical methods FBP and TV are highly transferrable across methods significantly reducing the reconstruction quality. The adversarial examples of neural network methods FBP-Unet, iRadonMap and LearnedGD are also transferrable to other network based approaches. The adversarial examples of LearnedPD are least transferable to other methods.

In addition to input specific adversarial examples, we also study the transferability of input-agnostic universal perturbations across different CT recovery methods. Table 5 summarizes the results of such transferability test for $\epsilon$ value of 0.05. The results indicate that even universal adversarial perturbations are transferable across different methods. This indicates the possibility of crafting fully black box attacks on CT recovery. The universal perturbation optimized for FBP is the most transferable to other methods, on the other hand universal perturbation optimized for LearnedPD is least transferable to other methods.

| Source Noise | FBP | FBP-Unet | iRadonMap | LearnedGD | LearnedPD |
|---|---|---|---|---|---|
| Clean | 30.53/0.714 | 35.67/0.824 | 34.19/0.799 | 34.74/ 0.802 | 35.92/0.829 |
| FBP | **10.34/0.036** | 9.90/0.031 | 8.74/0.025 | 7.68/0.021 | 10.62/0.041 |
| FBP-Unet | 14.42/0.098 | **4.95/0.022** | 9.06/0.035 | 9.26/ 0.095 | 7.77/0.042 |
| iRadonMap | 13.02/0.0706 | 9.61/0.049 | **3.82/0.0108** | 7.38/0.042 | 10.99/0.057 |
| LearnedGD | 15.60/0.188 | 13.52/0.220 | 10.38/0.112 | **4.32/0.183** | 9.69/0.109 |
| LearnedPD | 23.07/0.358 | 21.42/0.444 | 19.45/0.232 | 23.54/0.453 | **-2.95/0.003** |

Table 5: Evaluating transferability of universal adversarial noises for $\epsilon$=0.05.

| Method | $\hat{u}$ PSNR/SSIM | $(A\hat{u}, f)$ PSNR | $\epsilon$ | $\hat{u}_\delta$ PSNR/SSIM | $(A\hat{u}_\delta, f)$ PSNR | $(A\hat{u}_\delta, f_\delta)$ PSNR | $(f, f_\delta)$ PSNR | $\|\delta\|^2$ | $L_b$ Empir |
|---|---|---|---|---|---|---|---|---|---|
| FBP | 28.38/0.649 | 34.14 | 0.01 | 25.26/0.465 | 33.69 | 33.69 | 40.03 | 0.093 | |
| | | | 0.025 | 19.77/0.233 | 31.84 | 31.75 | 32.09 | 0.581 | 29.69 |
| | | | 0.05 | 14.36/0.096 | 28.78 | 28.52 | 26.12 | 2.292 | |
| TV | 28.94/0.652 | 37.47 | 0.01 | 24.88/0.520 | 36.58 | 36.54 | 40.10 | 0.092 | |
| | | | 0.025 | 18.91/0.302 | 33.32 | 33.74 | 32.20 | 0.565 | 33.98 |
| | | | 0.05 | 13.72/0.126 | 29.13 | 30.16 | 26.33 | 2.177 | |
| FBP-Unet | 33.55/0.799 | 36.50 | 0.01 | 19.37/0.384 | 34.52 | 35.244 | 40.14 | 0.091 | |
| | | | 0.025 | 12.82/0.115 | 28.33 | 29.23 | 32.31 | 0.551 | 97.56 |
| | | | 0.05 | 8.38/0.036 | 23.26 | 23.97 | 26.52 | 2.074 | |
| iRadonMap | 32.39/0.778 | 36.3 | 0.01 | 18.46/0.546 | 30.22 | 30.58 | 40.08 | 0.092 | |
| | | | 0.025 | 9.40/0.231 | 19.55 | 19.88 | 32.27 | 0.554 | 125.21 |
| | | | 0.05 | 5.39/0.051 | 14.92 | 15.12 | 26.65 | 2.01 | |
| LearnedPD | 33.64/0.802 | 36.50 | 0.01 | 17.75/0.412 | 34.23 | 34.92 | 40.11 | 0.092 | |
| | | | 0.025 | 10.56/0.153 | 31.26 | 33.08 | 32.34 | 0.548 | 108.48 |
| | | | 0.05 | 5.94/0.053 | 31.91 | 33.66 | 26.57 | 2.047 | |
| LearnedGD | 32.49/0.776 | 36.46 | 0.01 | 22.44/0.583 | 35.41 | 35.67 | 40.35 | 0.086 | |
| | | | 0.025 | 15.66/0.418 | 32.09 | 33.01 | 32.72 | 0.499 | 61.95 |
| | | | 0.05 | 10.89/0.301 | 29.10 | 30.02 | 27.19 | 1.773 | |

Table 6: Comparison of robustness to untargeted attacks on different CT reconstruction methods using 20 attack iterations on 100 samples LoDoPAB200 testset.

## Appendix C. Additional Results

**Untargeted Attacks on LoDoPAB_200**  Table 6 summarizes the results of our untargeted attacks on LoDoPAB_200 dataset, where the measurements are generating using 200 projection beams. Similar to our results on the LoDoPAB dataset, we find that classical approaches are more robust to untargeted attacks. However, on this dataset, the fully learned approach of iRadon Map is the most unstable method, followed by LearnedPD. LearnedGD is stable among the network based methods. Further, the methods show a trend of have a higher value of $L_b$ on LoDoPAB_200 dataset in comparison with LoDoPAB dataset indicating higher instabilities as the reconstruction from 200 projection beams is more severely ill-posed than from 513 projections.

**Qualitative Results**  Figure 3 shows the results of untargeted attack on two example CT images for three adversarial noise levels. The clean and adversarial reconstructions for the methods are shown. The visual results also indicate relative robustness of classical approaches to untargeted attacks.

Figure 4 shows result of localized attack on an example CT image for adversarial noise level of 0.01. The adversarial noise that produces the localized changes is also depicted. We can observe that the attack successfully modifies the local region using extremely low

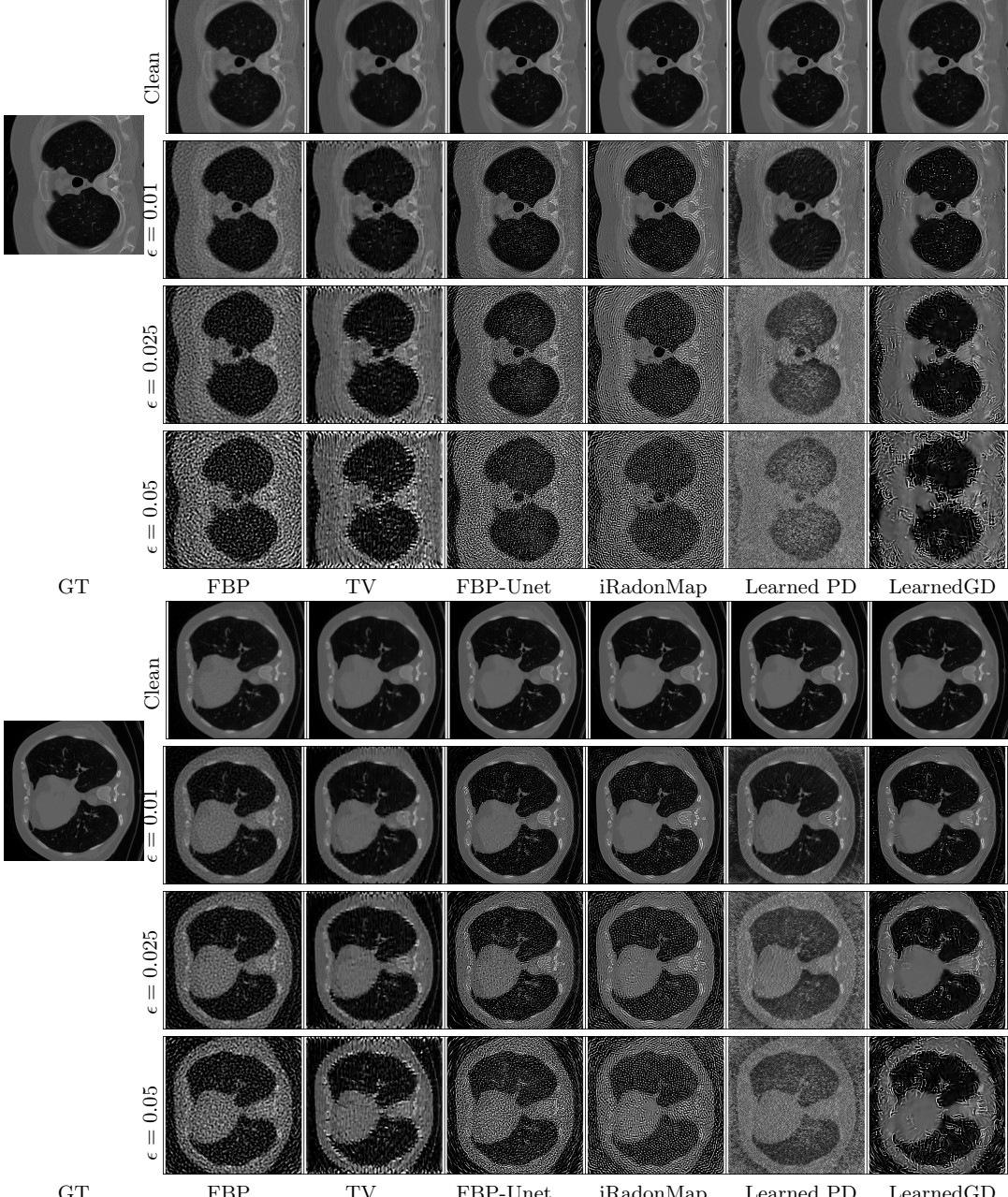

Figure 3: Untargeted attack on CT reconstruction methods for $\epsilon$ values 0.01, 0.025 and 0.05.

noise level. Figure 5 shows the results of localized attacks on 20 example CT images in LoDoPAB test set. For each method, the local patches extracted from clean and adversarial reconstructions are shown.

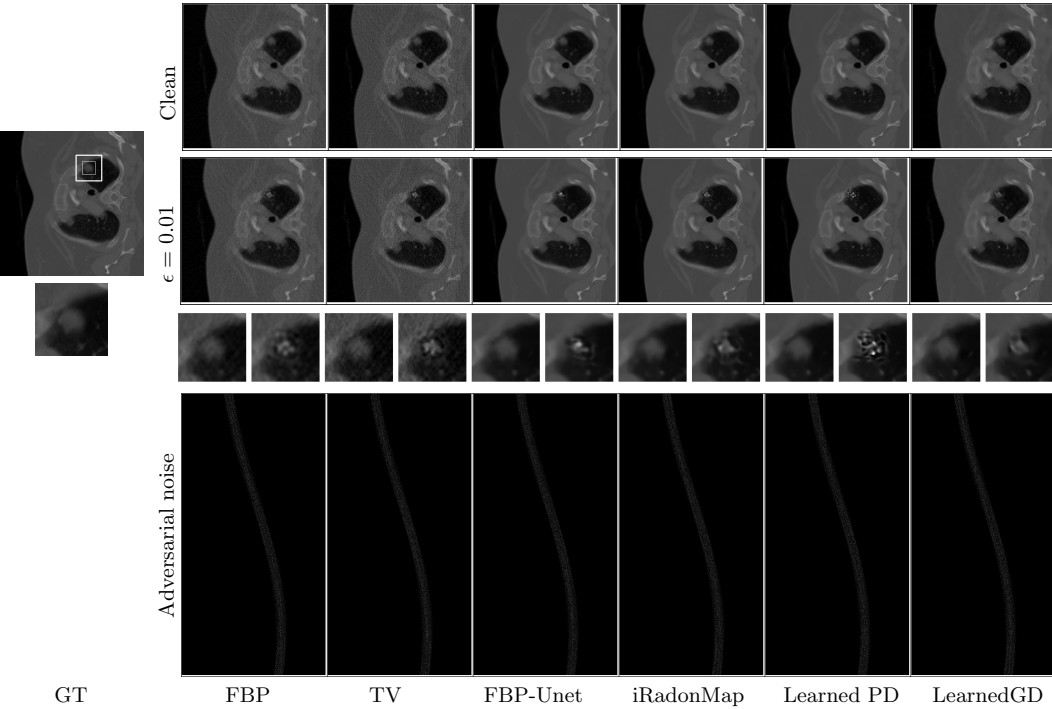

Figure 4: Localized attack on CT reconstruction methods. for $\epsilon = 0.01$. First and second row illustrate clean and adversarial reconstructions for each method. The third row shows the cropped patches from the clean (left) and adversarial (right) reconstructions. Adversarial noise in the fourth row is multiplied by $\times 25$ for visibility.

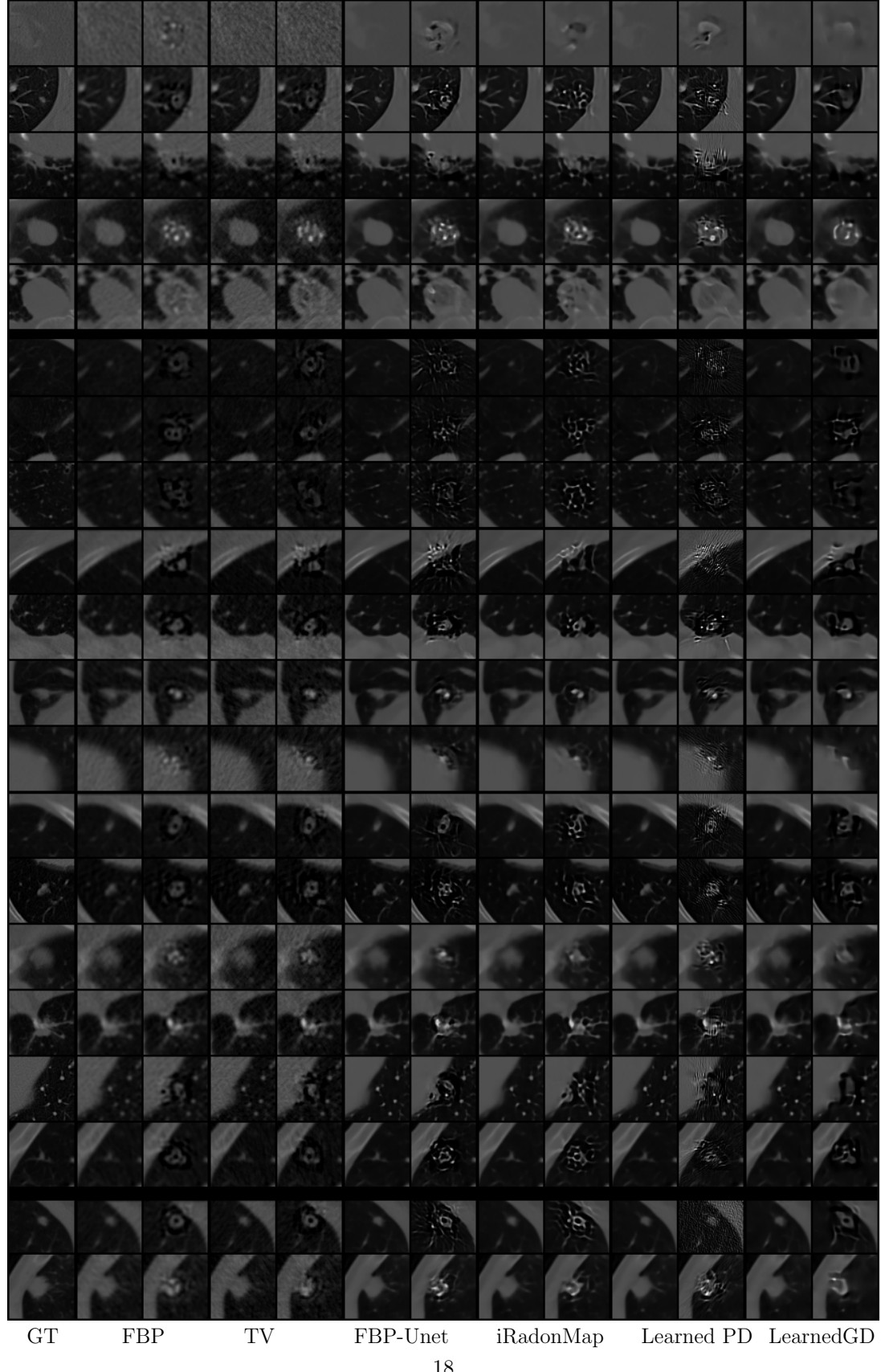

GT    FBP    TV    FBP-Unet    iRadonMap    Learned PD   LearnedGD

Figure 5: Result of localized attacks on 20 images. For each method left patch is from clean reconstruction and right is the result of attack.

