# OpenReview forum: "Evaluating Adversarial Robustness of Low dose CT Recovery"
_MIDL.io/2023/Conference — MIDL 2023 Poster_

### Official Review · Reviewer_1a2Y · 2023-01-31

**Confidence:** 3
**Preliminary Rating:** 3

**Summary:**

This work focuses on the robustness of CT reconstructions algorithms to adversarial attacks. They evaluate the performance of several reconstruction algorithms, including classical methods and deep networks, under two types of adversarial attacks: untargeted and localized attacks. The results suggest that deep learning based reconstruction algorithms are more susceptible to the untargeted attacks than classical ones. Furthermore, both deep learning based and classical algorithms are easily affected by localized attacks highlighting the need of better regularisation.

**Strengths:**

- The paper is written clearly and easy to understand.
- This discover a phenomenon that it is easy to attack reconstruction algorithms such that the resulting CT reconstructions change malignancy prediction evaluated by a pre-trained predictor.
- The localised adversarial attack is interesting,

**Weaknesses:**

- The technical contribution of this paper is limited. The fact that deep learning reconstruction methods are susceptible to adversarial attacks is known and has been actively researched. It could be better to point out some potential solutions as well.


**Deanonymize Review:**

no

**Detailed Comments:**

- I believe better figure captions and better presentation of figures will be helpful.


**Paper Type:**

validation/application paper

**Questions To Address In The Rebuttal:**

- When performing localized attacks, what does the adversarial noise added to f look like? Do they have some special patterns? How are they different from the adversarial noise for untargeted noise?

- The evaluation results show that for all methods, localised attacks lead to a 100% success rate. Do you think it is possible that the attack adds some ‘shortcut' artefacts (specific to this classifier) to the reconstruction, which leads to the 100% success rate? I am curious to see if you use different pre-trained classifiers for adversarial attack and evaluation, what the results will look like.

- It would be great if the authors can point out some potential solutions to help improve the robustness.

---

### Official Review · Reviewer_jj8t · 2023-02-04

**Confidence:** 5
**Preliminary Rating:** 4
**Recommendation:** Poster

**Summary:**

The paper focuses on the robustness against adversarial attacks for classic and neural network-based image formation approaches used in low-dose CT imaging. The authors consider three types of neural network-based image formation approaches, including i) models that directly predict images from undersampled sinograms, ii) models that denoise back-projected images, and iii) unrolled optimization networks, and compare their robustness against untargeted global and local attacks. For the classical approach, filtered back-projection with TV regularized is used.  They consider white box attacks where the attacker has complete knowledge of the image formation network and the classifier network. Projected gradient descent (PGD) with different levels of perturbations is used for attacking sinograms, and the authors compare the robustness of the neural network-based attack with the classical approach. They conclude that NN-based solutions are more brittle against untargeted attacks, while both classical and NN-based approaches are brittle against localized attacks (which is due to the brittle nature of the NN-classifier!).

**Strengths:**

* The paper is well written
* Comparison of the robustness of classical approaches against deep neural network-based solutions could be of interest to researchers
* The experiments support the claims of the paper

**Weaknesses:**

Main concern:

* Studying the vulnerabilities of NN-based image formation approaches demonstrates a worst-case behavior of these networks. However, the proposed attack has little to no physical-world realization. Here is why:
  1. The attacker has full knowledge of both image formation and classifier modules,
  2. The attacker has access to the raw sinogram, which means that the attacker can insert noise in the middle of the image formation pipeline,
  3. The attacker has time to optimize an image-specific adversarial perturbation.
These assumptions make the proposed approach trivial and of little interest to physical-world imaging.

* The paper contains no novelty. In particular, there are no specific considerations required for attacking the sinograms, and this has been done in prior publications, for instance, see: "Wu, W., Hu, D., Cong, W., Shan, H., Wang, S., Niu, C., ... & Wang, G. (2022). Stabilizing deep tomographic reconstruction: Part B. Convergence analysis and adversarial attacks. Patterns, 3(5), 100475." Hence, the only novelty is the comparison of the classical approach against NN-based approach.





**Deanonymize Review:**

no

**Paper Type:**

validation/application paper

**Questions To Address In The Rebuttal:**

Please see my comments in the Weaknesses. In addition, to make the proposed attack more interesting, I suggest that the authors study:

1) universal sinogram perturbations - designing a single perturbation will at least make the attack more realistic.
2) black-box attacks

The authors already provide preliminary evidence that the adversarial perturbations are transferable; therefore, it would have been great if they could provide rigorous studies for black-box and universal perturbation attacks.

---

### Official Review · Reviewer_94TQ · 2023-02-06

**Confidence:** 3
**Preliminary Rating:** 3

**Summary:**

This work compared the robustness of CT reconstruction methods when adversarial attacks/noise are presented. 6 CT reconstruction methods including traditional methods such as FBP and deep learning-based methods are evaluated on the low-dose parallel beam (LoDoPaB) CT dataset. The authors considered two types of adversarial attacks: untargeted attacks and localized attacks. Experimental results show that deep learning-based methods are more prone to untargeted attacks, but both traditional and deep learning methods are affected by localized attacks.

**Strengths:**

+ The performance metrics are well-designed to evaluate different methods comprehensively.
+ It is interesting that the localized attacks affect both classical approaches and deep learning approaches, but hardly change the data consistency.

**Weaknesses:**

- It is not clear what criteria are used to select the compared methods, especially deep learning approaches (see "Questions To Address In The Rebuttal" for details).
- The manuscript, particularly Section 4, is somewhat cumbersome and hard to digest.

**Deanonymize Review:**

no

**Detailed Comments:**

Table 1 and 2 are too dense to get any meaningful conclusion quickly. It would be very helpful to simply/redesign the presentation in a more efficient way.

**Paper Type:**

validation/application paper

**Questions To Address In The Rebuttal:**

Criteria of Method Selection
1. It is unclear how the 6 methods, especially the 4 deep learning approaches, are selected. Specifically, why the selected methods are comprehensive enough to represent category i)-iii) in Section 2.2.
2. It is well known that deep learning is prone to adversarial attacks, which has also been studied in medical image recovery applications (Genzel et al., 2022, etc.) Therefore, it is less significant to evaluate them on one CT reconstruction dataset, especially for untargeted attacks. Instead, it may be more interesting to evaluate deep learning methods designed to be robust to adversarial attacks/noises.

---

### Meta-Review · Area_Chair_iamy · 2023-02-20

**Recommendation:** Accept (Poster)
**Confidence:** 4

**Metareview:**

After reading the reviews, and authors responses, I believe the authors have generally addressed most important concerns raised by the reviewers. From the discussions, I believe that this work can lead to interesting in-person discussions/debate regarding the transferability of the adversarial attacks and the practical significance of localized attacks. Thus, I recommend the acceptance of this submission.